

# Cyclodipeptides from *Pseudomonas aeruginosa* modulate the maize (*Zea mays* L.) root system and promote S6 ribosomal protein kinase activation

Iván Corona-Sánchez[1], Cesar Arturo Peña-Uribe[1],
Omar González-López[1,2], Javier Villegas[3], Jesus Campos-Garcia[2] and
Homero Reyes de la Cruz[1]

[1] Instituto de Investigaciones Químico Biológicas, Laboratorio de Biotecnología Molecular de Plantas, Universidad Michoacana de San Nicolás de Hidalgo, Morelia, Michoacán, México
[2] Instituto de Investigaciones Químico Biológicas, Laboratorio de Biotecnología Microbiana, Universidad Michoacana de San Nicolás de Hidalgo, Morelia, Michoacán, México
[3] Instituto de Investigaciones Químico Biológicas, Laboratorio de Agroecología, Universidad Michoacana de San Nicolás de Hidalgo, Morelia, Michoacán, México

Corresponding author
Homero Reyes de la Cruz,
delacruz@umich.mx

## ABSTRACT

**Background:** *Pseudomonas aeruginosa* is an opportunistic and pathogenic bacterium with the ability to produce cyclodipeptides (CDPs), which belong to a large family of molecules with important biological activities. Excessive amounts of CDPs produced by *Pseudomonas* strains can activate an auxin response in *Arabidopsis thaliana* and promote plant growth. Target of rapamycin (TOR) is an evolutionarily conserved eukaryotic protein kinase that coordinates cell growth and metabolic processes in response to environmental and nutritional signals. Target of rapamycin kinase phosphorylates various substrates, of which S6 ribosomal protein kinase (S6K) is particularly well known. The PI3K/Akt/mTOR/S6K signaling pathway has been studied extensively in mammals because of its association with fundamental biological processes including cell differentiation. However, evidences suggest that this pathway also has specific and conserved functions in plants and may thus be conserved, as are several of its components like TOR complex 1 and S6K proteins. In plants, TOR-S6K signaling has been shown to be modulated in response to plant growth promoters or stressors.

**Methods:** In this study, we evaluated the effects of *P. aeruginosa* CDPs on the growth and root development of maize plants (*Zea mays* L.) by adding different CDPs concentrations on culture plant media, as well as the effect on the phosphorylation of the maize S6K protein (*Zm*S6K) by protein electrophoresis and western blot.

**Results:** Our results showed that *P. aeruginosa* CDPs promoted maize growth and development, including modifications in the root system architecture, correlating with the increased *Zm*S6K phosphorylation and changes induced in electrophoretic mobility, suggesting post-translational modifications on *Zm*S6K. These findings suggest that the plant growth-promoting effect of the *Pseudomonas* genus, associated with the CDPs production, involves the TOR/S6K signaling pathway as a mechanism of plant growth and root development in plant–microorganism interaction.

# INTRODUCTION

Living organisms can typically adjust their metabolism in response to resource availability in their environment. Favorable environmental stimuli (availability of nutrients, energy, or growth factors) promote anabolic processes in cells, which leads to an increase in size and biomass. In contrast, exposure to stressors such as nutrient or energy deficits, hypoxia, or DNA damage leads to the downregulation of macromolecule synthesis and upregulation of catabolic processes like autophagy (*Wullschleger, Loewith & Hall, 2006*).

Inoculation of *Arabidopsis thaliana* plants with a *Pseudomonas aeruginosa lasI/rhlI* double mutant strain substantially increased shoot and root biomass production. This mutant strain lacks both *lasI* and *rhlI* genes, therefore is unable to produce N-acyl-L-homoserine lactones, which are *quorum sensing* molecules that regulate virulence factors production. The compounds responsible for this increase in plant growth were identified as cyclic dipeptides (cyclodipeptides (CDPs)): cyclo(L-Pro-L-Tyr), cyclo(L-Pro-L-Val), and cyclo(L-Pro-L-Phe), which also activated the auxin-inducible DR5:*uidA* marker in transgenic *Arabidopsis* plants (*Ortiz-Castro et al., 2011*). Cyclodipeptides are non-ribosomal peptides synthesized by a large number of organisms and constitute a novel family of heterocyclic compounds with different important biological activities (*Belin et al., 2012*; *Hernández-Padilla et al., 2017*). Cyclodipeptides have been shown to affect plant development by eliciting various responses including growth promotion (*Cronan et al., 1998*; *Degrassi et al., 2002*; *Kimura et al., 2005*), germination and hydric stress resistance (*Ienaga et al., 1990*), expression of pathogen resistance genes (*Park et al., 2016*), and promotion of lateral root formation (*Ortiz-Castro et al., 2011*; *González et al., 2017*; *Ortiz-Castro, Campos-García & López-Bucio, 2019*).

Target of Rapamycin (TOR) protein kinase is a key cell growth regulator in eukaryotes and it is important for cell growth control and metabolic responses to environmental signals. The genomes of most examined eukaryotes contain a single copy of the TOR gene and knockout studies indicate that the mutation of this gene is lethal, inducing developmental arrest at an early stage (*Menand et al., 2002*; *Gangloff et al., 2004*; *Xiong et al., 2016*). In mammals, TOR forms part of two complexes referred to as mTORC1 and mTORC2 that differ in functions, subunit composition, and susceptibility to rapamycin, an antifungal macrolide with strong immunosuppressive and antiproliferative properties (*Vézina, Kudelski & Sehgal, 1975*; *Martel, Klicius & Galet, 1977*; *Eng, Sehga & Vézina, 1984*). The mTORC2 complex (comprising the TOR, RICTOR, and mLST8 subunits) is not inhibited by rapamycin and its function is associated mainly with actin cytoskeleton polarization and endocytosis (*Kamada et al., 2005*; *Niles & Powers, 2014*; *Rispal et al., 2015*). In contrast, the mTORC1 complex is susceptible to rapamycin inhibition and forms part of a signaling pathway that upregulates anabolic processes and suppresses catabolic processes when the environmental conditions are favorable for growth.

The mTORC1 complex regulates cell growth in response to four main factors: insulin, insulin-like growth factors (IGF), nutrients, energy levels, and stressors (e.g., lack of

nutrients or energy, hypoxia, or DNA damage). Eukaryotic translation initiation factor 4E (eIF4E)-binding protein (4EBP) and the S6 protein kinase (S6K), which phosphorylates the S6 ribosomal protein (a component of the small ribosomal subunit) were the first metazoan targets of mTORC1 to be identified (*Meyuhas & Dreazen, 2009*; *Fenton & Gout, 2011*). Activation of S6K by mTOR is essential for overall protein synthesis efficiency and cell size control (*Meyuhas, 2015*) and modulates transcription, lipid synthesis, cell growth and size, and cellular metabolism (*Magnuson, Ekim & Fingar, 2012*). Compared to animals and yeasts, TOR signaling has not been extensively studied in plants; however, conserved components of this signaling pathway have been identified in plants and were found to exert functions similar to those of their mammals and yeasts counterparts (*Dobrenel et al., 2016a*; *Schepetilnikov & Ryabova, 2018*; *Bakshi et al., 2019*).

In maize, the presence of TOR-S6K signaling pathway components has been confirmed, including a TOR kinase protein (*Zm*TOR) (*Agredano-Moreno et al., 2007*), an insulin-like growth factor (*Zm*IGF) (*García Flores et al., 2001*), and an S6K protein kinase (*Zm*S6K) ortholog of the human p70S6K isoform (*Reyes de la Cruz, Aguilar & Sánchez de Jiménez, 2004*). Maize S6K comprises two of the main target residues that are essential for p70S6K activation in animals, namely Thr-229 (Ser-308 in *Zm*S6K) and Thr-389 (Thr-468 in *Zm*S6K), which is phosphorylated by TOR kinase and is critical for S6K activation (*Wullschleger, Loewith & Hall, 2006*).

In plants, TOR signaling has been shown to be activated by auxins and exposure to light, as well as glucose and amino acids, likely because a TOR signaling pathway evolved in plants as a consequence of sedentariness and photo-autotrophy (*Dobrenel et al., 2016b*; *Schepetilnikov & Ryabova, 2018*; *Bakshi et al., 2019*). *Pseudomonas aeruginosa* PAO1 CDPs can modulate several aspects of *A. thaliana* development (*Ortiz-Castro et al., 2011*; *González et al., 2017*); thus, in the present study, we assessed the effects of CDPs on growth and development of maize plants (*Zea mays* L.). Furthermore, we examined whether CDPs treatment induces the activation of the TOR signaling pathway through stimulation of S6K kinase phosphorylation.

# MATERIALS AND METHODS

## Plants and growth conditions

*Zea mays* L. 'Chalqueño' seeds were used in all the experiments. The seeds were surface-sterilized using 50% (v/v) sodium hypochlorite (NaClO, 6% active Cl) for 5 min and washed three times with deionized water. The seeds were then treated with 70% (v/v) ethanol for 5 min and again washed three times with deionized water. After this, the seeds were placed in cotton beds soaked with sterile water for germination (72 h, 25 ± 2 °C, in darkness). Seedlings that showed homogeneous growth were selected, and the endosperm was removed using a scalpel. Seedlings were surface-sterilized with sodium hypochlorite and ethanol, as detailed above, and placed in $25 \times 150$ mm tubes containing five mL of liquid MS medium (*Murashige & Skoog, 1962*) according to *Martínez-de la Cruz et al. (2015)*. Solutions containing CDPs at different concentrations (0.5, 1, 10, 20, and 50 µM) or 12 µM indole-3-acetic acid (IAA) were added to the tubes. The highest amount of DMSO used for preparing the different concentration of CDPs was used for control

treatments (1.5 µL DMSO/mL of medium) because CDPs stock was prepared in 80% DMSO. The tubes were placed in a plants growth chamber with a photoperiod of 16 h light and 8 h darkness at $22 \pm 2\,°C$ for 8 days. After this, plant growth was assessed by recording the shoot length, primary root length, number of lateral roots on the primary root, number of seminal roots, and number of crown roots.

To evaluate the effect of CDPs on maize plants, maize seeds were surface-sterilized as outlined above and germinated for 24 h. Seeds with homogeneous germination were selected, placed in pots with sterilized soil, and grown for 30 days. The seedlings were irrigated every third day. The control group received sterilized water, and the CDPs treatment group received sterile water containing CDPs at a concentration of 20 µM. The *P. aeruginosa* CDPs mixture used in the different treatments were obtained as described by *Ortiz-Castro et al. (2011)* and *González et al. (2017)*.

## Plant growth analyses

The foliar and primary root lengths of plants were measured using a ruler. The numbers of lateral roots on primary, seminal, and crown roots were counted manually, and total root length, number of lateral roots, average root diameter, root volume, and surface area were recorded by automated scanning using an Epson Expression 11000XL A3 flatbed photo scanner and WhinRHIZO root imaging software (Regent Instruments Inc., Chemin Sainte-Foy, Quebec, Canada). Quantitative data were analyzed with GraphPad Prism 5 (GraphPad Software Inc., La Jolla, CA, USA). A one-way ANOVA was performed followed by a Dunnett's *post-hoc* test and a Student's *t*-test for independent samples. The results are expressed as the arithmetic means ± standard error.

## Protein extraction

Embryonic axes from seeds germinated for 22 h were dissected and incubated in flasks containing either MS medium only or MS medium supplemented with CDPs at concentrations of 0.5, 1, 10, 20, or 50 µM, or with 12 µM IAA. The flasks were kept with slight agitation (120 rpm) for 2 h, and the embryonic axes were subsequently frozen using liquid nitrogen and then ground using a mortar. After this, an extraction buffer (50 mM HEPES, 1 mM sodium orthovanadate, 1 mM sodium molybdate, 1 mM EDTA, 1 mM benzamidine, 20 mM NaF, 80 mM beta-sodium glycerophosphate, 0.2 mM PMSF, and 2 mM DTT at pH 7.6) was added to the sample powder and the mixture was centrifuged at $9,000 \times g$ at $4\,°C$ for 30 min. The protein content was determined on supernatants by the Bradford method (*Bradford, 1976*).

## Protein gel blot analyses

Protein extract samples (30 µg) were separated by SDS–PAGE (12% acrylamide) and blotted on PVDF membranes (Millipore, Burlington, MA, USA). The membranes were incubated with either an anti-human-phospho-p70S6K-Thr389 (p-p70S6K) polyclonal antibody (1:1,000 dilution) (Santa Cruz Biotechnology, Dallas, TX, USA), or with an anti-ZmS6K polyclonal antibody (1:1,000 dilution) (Abbiotec, San Diego, CA, USA) at $4\,°C$ overnight. Membranes were washed with 0.15 M TBST for 10 min, then with 1 M TBST for

10 min, and then twice with 0.15 M TBST for 10 min. After this, the membranes were incubated for 2 h with a peroxidase-conjugated goat anti-rabbit IgG antibody (1:3,000 dilution) (Bio-Rad Laboratories, Inc., Hercules, CA, USA) at room temperature. The membranes were washed five times with 0.15 M TBST for 5 min each. The peroxidase reaction was revealed using luminol and detected using a ChemiDoc™ XRS+ system (Bio-Rad Laboratories, Inc., Hercules, CA, USA). Densitometry analysis for each protein blot was performed using ImageJ software (NIH, Bethesda, MA, USA).

## 2D gel blot

For 2D gel blot assays, the 8-day-old maize plants were dissected in two parts: root and shoot, of which only the root was used because obvious changes were observed in this area by CDPs treatment. Tissues were ground with liquid nitrogen and the powder placed in extraction buffer (50% phenol [v/v], 0.9 M sucrose, 10 mM EDTA, 0.4% 2-mercaptoethanol [v/v], 100 mM Tris) with an equal part of phenol solution (equilibrated with 10 mM Tris HCl, pH 8.0, 1mM EDTA), homogenized in vortex and centrifuged at 9,000×$g$ at 4 °C for 10 min. The organic phase was recovered and an equal amount of extraction buffer was added. The sample was homogenized and centrifuged at 9,000×$g$ at 4 °C for 10 min. The organic phase was recovered again and a precipitation solution (ammonium acetate/methanol) was added to precipitate the proteins at −20 °C overnight. After this period, the sample was centrifuged at 9,000×$g$ at 4 °C for 10 min, and the precipitate washed three times with precipitation solution and final wash with cold acetone at 80%, performing a centrifugation at 9,000×$g$ at 4 °C for 5 min between washes. The pellet was dissolved in isoelectric focusing buffer IEF (8M urea, 4% CHAPS, 7mM DTT, 2% ampholytes) and proteins were quantified by the Bradford method. Each protein sample (150 μg) was placed in 13 cm strips with an immobilized pH gradient of 3–10 (IPG DryStrip pH 3-10 NL,13 cm; GE Healthcare Life Sciences). The strips were left to hydrate with the samples overnight. Fist dimension separation of proteins was performed with "Ettan IPGphor 3 Isoelectric Focusing Unit". Subsequently, the strips were placed in 12% acrylamide gels for SDS–PAGE. Then, the gels were blotted on PVDF membranes (Millipore). Membranes were incubated with anti-ZmS6K (1:1,000 dilution), peroxidase-conjugated anti-rabbit antibody (1:3,000 dilution) and revealed as described above.

## RESULTS

### Modification of maize growth and development by *P. aeruginosa* CDPs

To evaluate the effect of CDPs on maize growth and development, we performed experiments using the *in vitro* system described in *Martínez-de la Cruz et al. (2015)*, except that in the step of growing embryonic axes of 24 h in agar-solidified MS medium and cultured for 7 days, in this work the seeds were allowed to germinate 72 h and the seedlings were surface disinfected and placed in liquid MS media supplemented with CDPs or IAA as described in the "Materials and Methods" section. It was previously shown that 12 μM of IAA modify the maize root development (*Martínez-de la Cruz et al., 2015*).

Maize seedlings treated with *P. aeruginosa* CDPs for 8 days showed no significant changes in shoot growth at any treatment concentration compared to both the control and

the 12 µM IAA-treated seedlings (Fig. 1). Shoot growth (shoot length and primary root length) was not significantly different between plants treated with CDPs or IAA and the controls (Figs. 2A and 2B). In contrast, 20 µM CDPs treatment elicited a substantial increase in root system development compared to the other CDPs concentrations, IAA treatment, or control. The 20 µM CDPs and 12 µM IAA treatments led to a significant increase in the number of lateral roots (Fig. 2C). Moreover, there was a significant increase in the number of seminal roots, but not crown roots, in 20 µM CDP-treated versus control plants (Figs. 2D and 2E, respectively).

The effect of CDPs on maize plants was assessed over longer exposure periods to determine whether the previously observed effects would also occur at more advanced developmental stages. For this, maize plants were grown for 30 days in pots and treated with 20 µM CDPs based on the results obtained previously with the 8-days treatment. After 30 days, the treated maize plants showed increased growth and development, evidenced by increased shoot length and root biomass as showed at 20 µM CDPs treatment (Fig. 3). All the parameters showed increasing trends in CDPs-treated plants (Fig. 4, except for root diameter); however, only the differences in shoot length (Fig. 4A), total root length (Fig. 4B), lateral roots number (Fig. 4C), root volume (Fig. 4E), and root surface area (Fig. 4F) showed significant statistical differences compared to the controls, whereas the differences in root diameter, where a slight decrease was observed, were not significant (Fig. 4D). These results indicate that at advanced plant development stages, the CDPs significantly induces shoot and root development.

## Effect of *P. aeruginosa* CDPs on S6K activation

The TOR kinase signaling pathway has been implicated in regulating growth and development in photo-synthetic and non-photosynthetic eukaryotes. To assess the association of the TOR pathway with the promotion of plant growth in CDPs-treated maize plants, protein gel blot analyses were performed using antibodies against anti-human p70-S6K phosphorylated at Thr389 (p-p70S6K), which is phosphorylated by TOR kinase and is critical for S6K activation. The results showed that CDPs treatment elicited a dose-dependent increase in *Zm*S6K phosphorylation (Fig. 5A), with the highest *Zm*S6K phosphorylation level occurring with 20 µM CDPs treatment, exceeding that of the 12 µM IAA treatment (Fig. 5B). Immunodetection with an antibody against *Zm*S6K showed that protein concentrations were not modified by either CDPs or IAA treatments (Fig. 5A). To further evaluate the effect of CDPs on *Zm*S6K activation, a 2D-PAGE protein gel blot analysis was performed using antibodies against *Zm*S6K. In our experimental conditions, two protein spots were detected at the molecular weight and pH range, corresponding to the *Zm*S6K protein. The results showed that there were changes in *Zm*S6K protein electrophoretic mobility in protein extracts from CDPs-treated plants; these changes could be observed as a displacement of protein spots towards more acidic pH (Fig. 6), indicating the appearance of *Zm*S6K isoforms with post-translational modifications, correlating with the increased levels of *Zm*S6K phosphorylation observed.

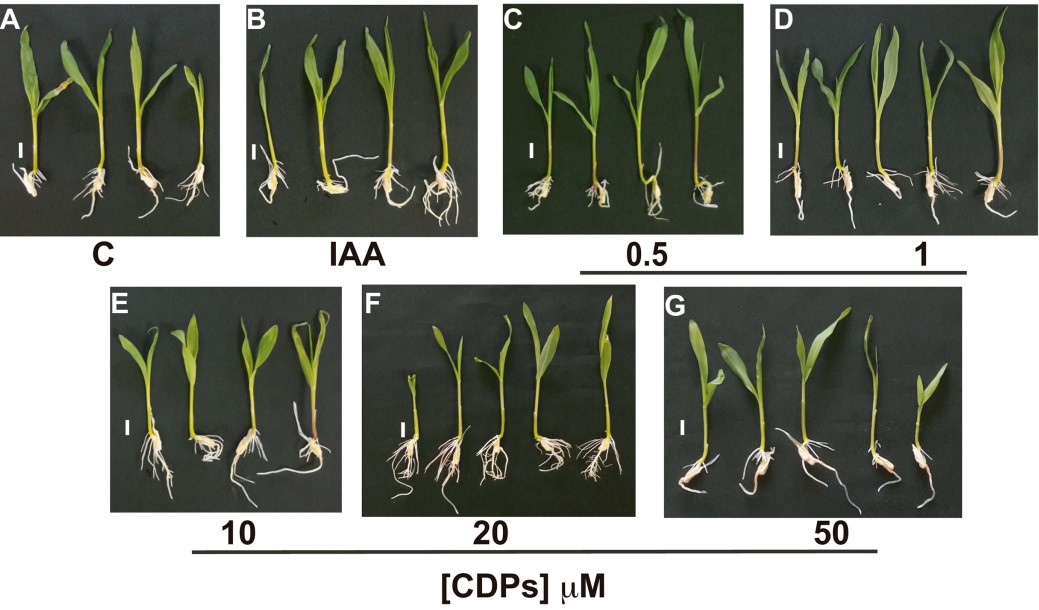

**Figure 1 Effect of cyclodipeptides (CDPs) on maize seedling growth.** Maize seedlings removed from seeds that germinated after 72 h were grown for 8 days in liquid MS medium. Control plants (A) and plants treated with 12 µM IAA (B) or CDPs at the indicated concentrations (C–G) are shown. Each group contained at least eight plants, and the experiments were repeated three times with similar results. Scale bar, one cm.

## DISCUSSION

We assessed the plant growth-promoting effects of CDPs produced by the *P. aeruginosa* PAO1 strain on maize plants. Maize plants treated with 20 µM CDPs for 8 days elicited a significant increase in lateral and seminal root formation (Figs. 1 and 2). As CDPs treatment significantly affected root development, but not shoot growth, the treatment period was extended to examine the effect of CDPs on the long-term growth and development of maize plants. Plants treated with CDPs (20 µM) for 30 days showed a slight increase in shoot growth (Figs. 3A and 4A) in addition to a significant increase in the number of lateral roots, root volume, and root surface (Figs. 3C, 4C, 4E, and 4F, respectively). These effects may have been due to the key role of this root type in early maize development since lateral roots provide stability and increase the soil contact and absorption area (*Grzesiak, 2009*).

Although the specific mechanism by which CDPs promote plant growth remain unknown, a mimicking of auxin signals was suggested and is supported by computational molecular docking analysis (*Ortiz-Castro et al., 2011*). This study indicated that CDPs molecules may fit in the *A. thaliana* auxin receptor binding pocket in a manner similar to the natural hormone IAA, although with different affinities, as observed for 1-naphthaleneacetic acid (NAA) and 2,4-dichloro-phenoxyacetic acid (*Ortiz-Castro et al., 2011*).

The promoting effect of CDPs on lateral roots is likely due to an auxinic response. Auxins are essential for lateral root formation in maize, as shown in studies using chemical auxin-transport inhibitors (*Jansen et al., 2012*). Moreover, it was shown that the highest

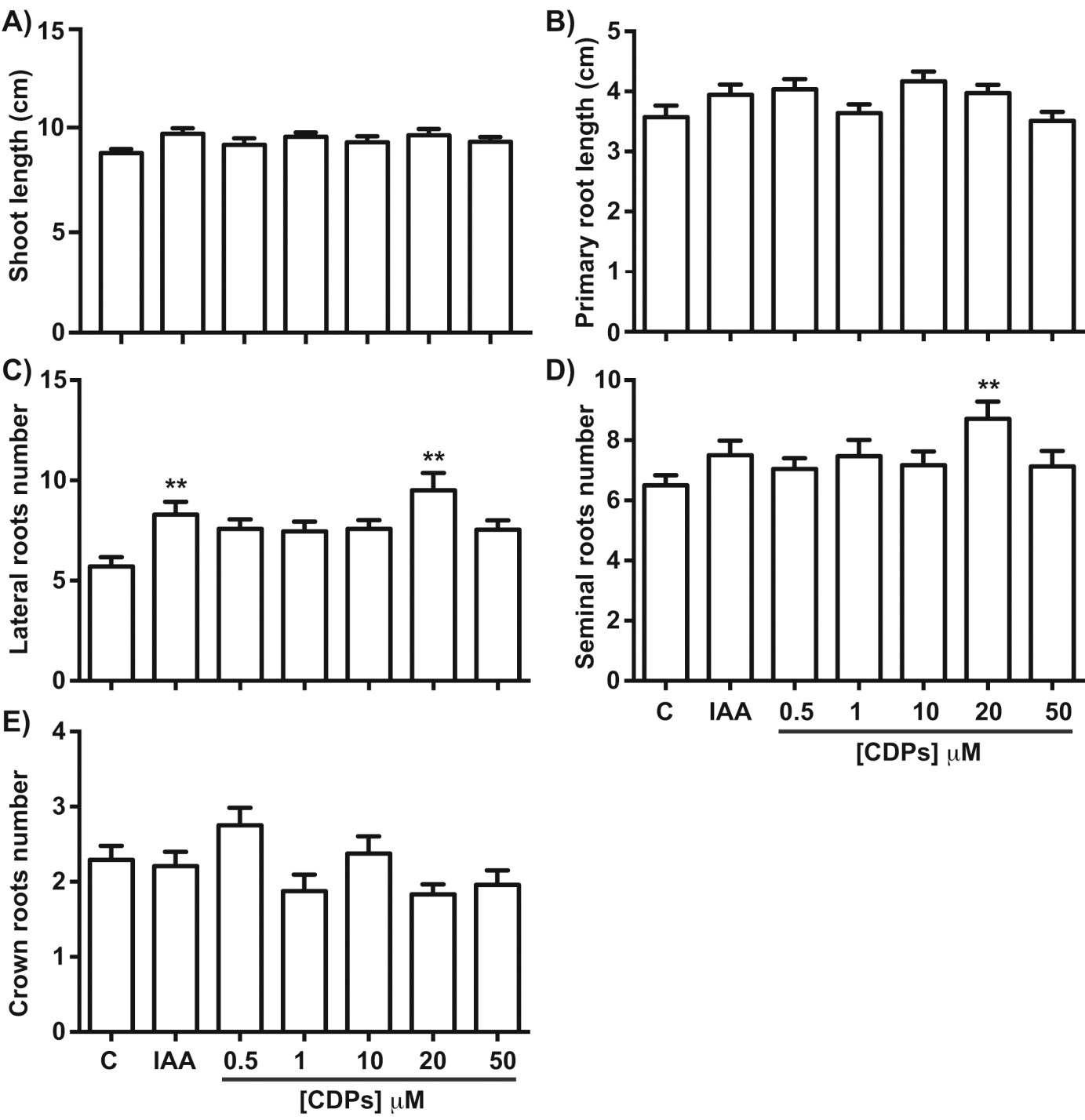

**Figure 2 Quantitative effect of cyclodipeptides (CDPs) on maize seedlings.** Measurements of different parameters in the plants shown in Fig. 1: (A) shoot length; (B) primary root length; (C) lateral root number of primary roots; (D) seminal root number; and (E) crown root number. The asterisks indicate statistically significant differences compared to the control treatment. Bars indicate means ± standard error ($n = 24$) from three independent experiments. One-way ANOVA, Dunnett's *post-hoc* multiple comparison test (**, $P < 0.01$).

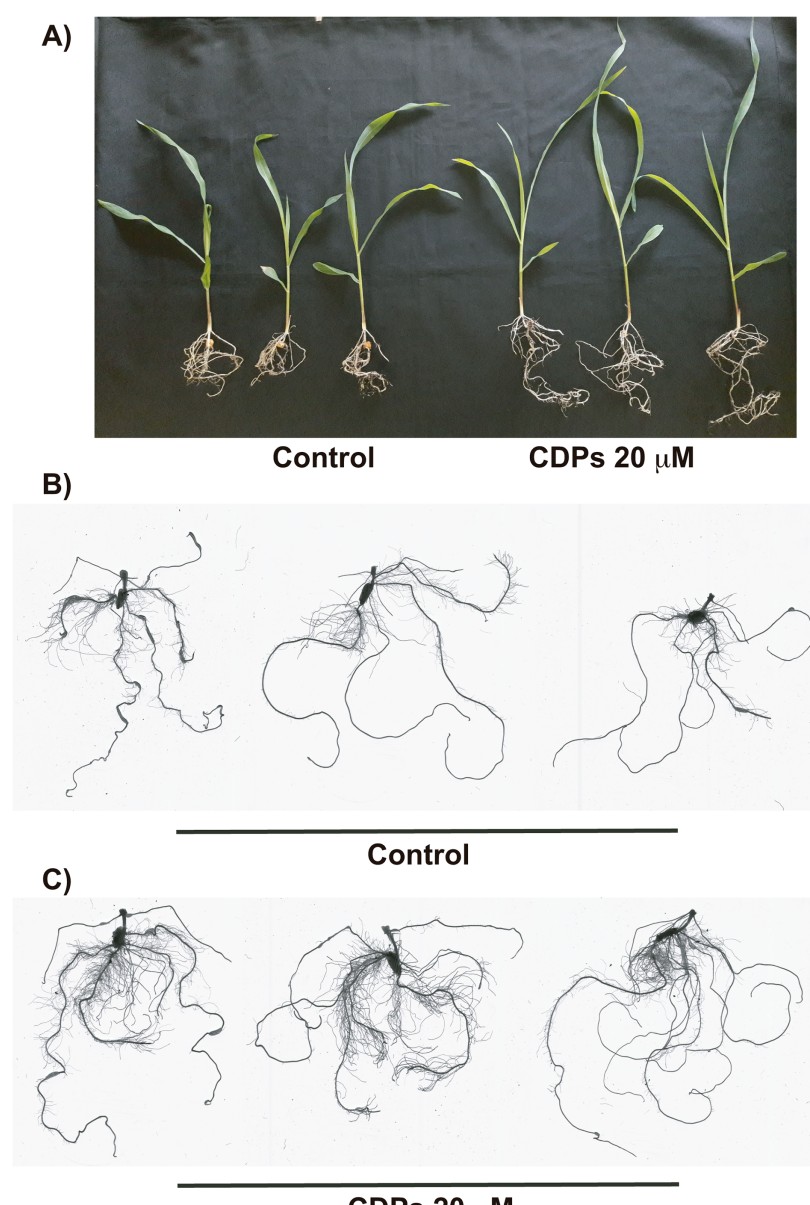

**Figure 3 Effect of cyclodipeptides (CDPs) on maize plants after 30 days.** The images show treated and control plants. (A) Representative plants for each treatment. (B) Representative root system images of control and (C) CDPs-treated plants.

auxin concentrations occur in the phloem, where auxin is crucial for lateral root formation; auxin deficit results in random divisions of pericycle and endodermis cells, which hamper the formation of new organs (*Jansen et al., 2012*).

Lateral roots are crucial for maize plant development within the first 2 weeks of growth. In this stage, lateral roots are the main sites of water absorption from the soil, whereas primary and seminal roots are responsible for water transport to the shoot (*Ahmed et al., 2016*).

The CDPs treatment induces notable root development on maize plants; however, at a concentration of 20 μM, CDPs treatment elicited statistically significant effects on maize

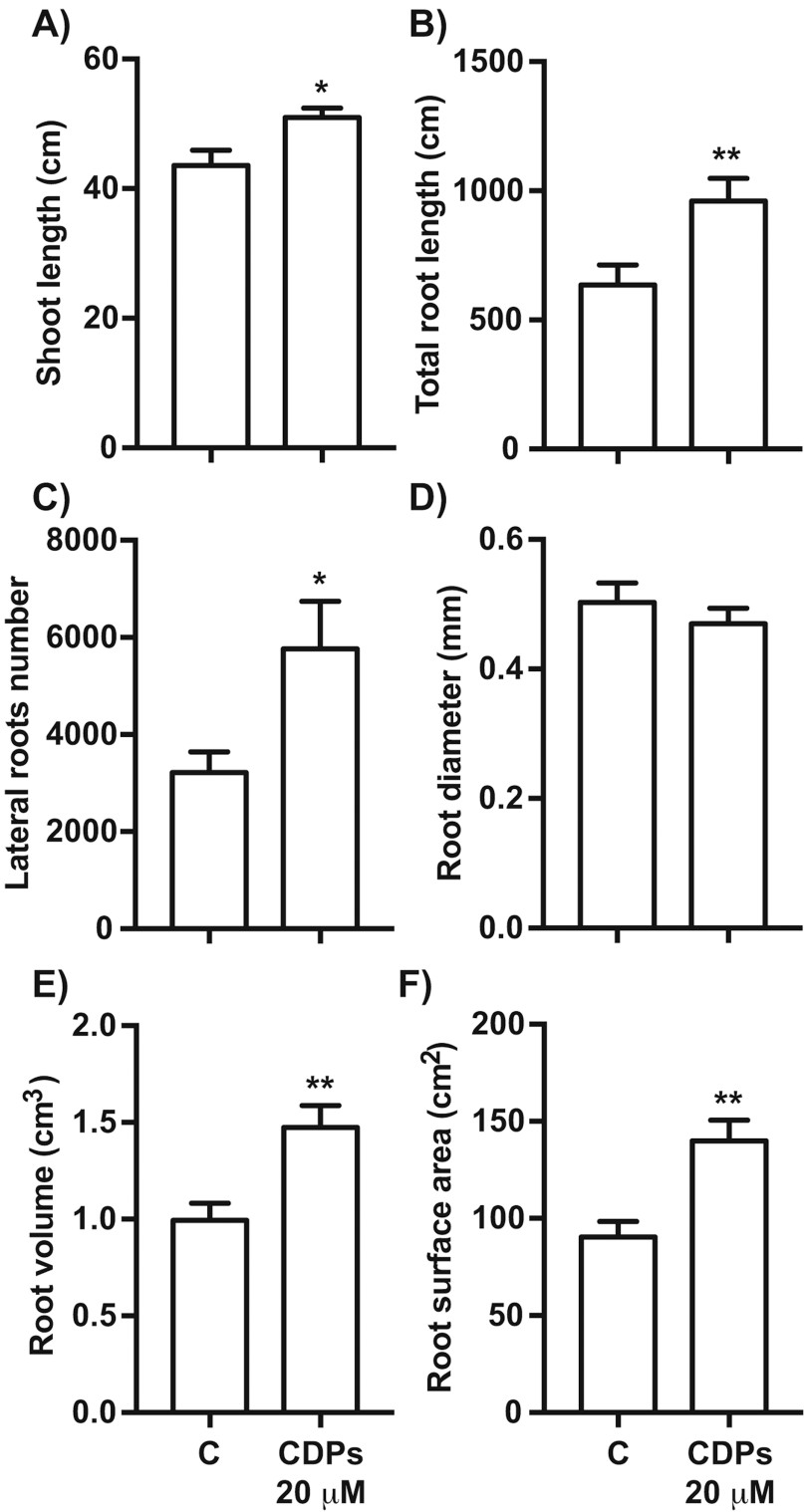

**Figure 4 Quantitative effect of cyclodipeptides (CDPs) on 30-day-old maize plants.** Measurements of morphological parameters in the plants shown in Fig. 3: (A) shoot length; (B) total root length; (C) lateral root number; (D) average root diameter; (E) root volume; and (F) root surface area. Asterisks indicate significant statistical differences compared to the control. Bars indicate means ± standard error of three experiments. Unpaired *t*-test (*, $P < 0.05$; **, $P < 0.01$).

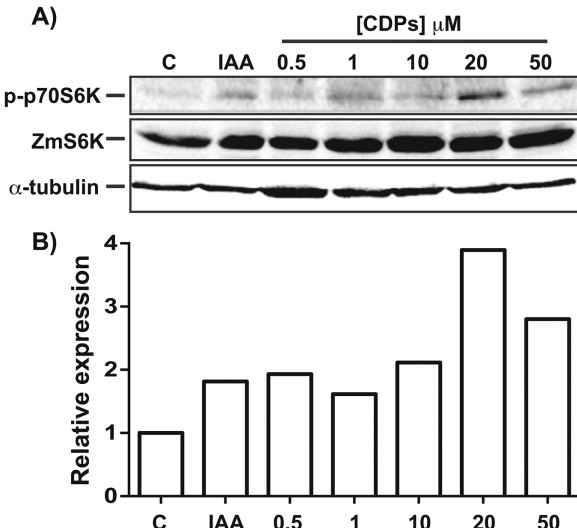

**Figure 5 Effect of *Pseudomonas aeruginosa* PAO1 cyclodipeptides (CDPs) on the phosphorylation of maize S6K.** Protein extracts from maize plants treated with 20 μM CDPs were separated by SDS–PAGE and analyzed by protein gel blot using the following antibodies: (A) anti-human p-p70S6K; anti-*Zm*S6K; and anti-human α-tubulin. The experiment was repeated twice with similar results; representative images are shown. (B) Densitometric analysis of representative membranes using anti-human p-p70S6K and anti-*Zm*S6K antibodies and normalized to α-tubulin detection.

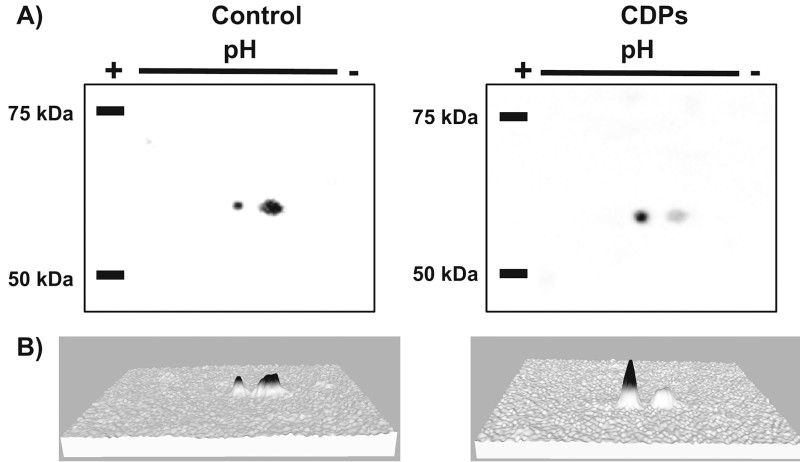

**Figure 6 Effect of *Pseudomonas aeruginosa* PAO1 cyclodipeptide (CDP) treatment on *Zm*S6K electrophoretic mobility.** The images show *Zm*S6K immunodetection by 2D-PAGE and protein gel blot analysis. (A) Shows representative images of immunodetected proteins from control and CDP-treated (20 mM) plants using an anti-*Zm*S6K antibody and detected with the ChemiDoc[TM] XRS+ system. (B) Shows 3D views of the images in (A) converted with Image Lab[TM] software (Bio-Rad Laboratories, Inc., Hercules, CA, USA). Protein standards, molecular mass in kDa, and the pH gradient are indicated. The experiment was repeated twice with similar results.

root architecture, which may have been due to a lower affinity of CDPs compared to auxins such as indole-butyric acid (IBA) and NAA, which exert effects at lower concentrations. Treatment of maize plants with various auxins has been shown to exert different dose-dependent effects on root architecture. At concentrations ranging from 1.5 to 3 μM,

IBA and NAA promoted lateral root formation and led to shortened primary roots, a phenotype associated with high concentrations of auxins; moreover, at concentrations ranging from 6 to 24 µM, IAA was shown to induce clear changes in root architecture (*Martínez-de la Cruz et al., 2015*).

As in lateral roots, the effects of CDPs on maize seminal root initiation may be due to an auxinic activity (Figs. 2–4). Seminal and lateral root numbers are determined by intrinsic developmental programs; however, previous studies indicated that initiation of seminal root formation was affected by several factors such as nutrient levels (i.e., growth in media with high or low phosphorus concentrations) (*Zhu et al., 2006*) and the effects of auxin-induced genes (*Taramino et al., 2007*).

Crown root formation was not significantly affected by CDPs treatment; however, it would be interesting to examine potential CDPs effects on adult plants with root systems dominated by crow roots.

Target of rapamycin kinase is the master regulator of cellular development by orchestrating nutrition/energy status with growth-induced signals. Studies on plants and mammals indicate that TOR only mediates the phosphorylation of RPS6 via S6K, which impact on the overall protein synthesis. S6 protein kinase phosphorylation at Thr388/389 is conserved both in animals and plants and is a widely used method for determining TOR activity (*Xiong & Sheen, 2012*; *Schepetilnikov et al., 2013*; *Ahn, Ahn & Pai, 2014*). Regarding the effect of CDPs on *Zm*S6K phosphorylation, the highest level of phosphorylation was observed at a CDPs concentration of 20 µM (Fig. 5). Interestingly, a stronger activation of *Zm*S6K was observed in the 20 µM CDPs treatment group than in the other groups, in agreement with the overall results for growth and development. Furthermore, *Zm*S6K phosphorylation levels decreased with 50 µM CDPs treatment, suggesting an inhibitory effect at higher CDPs concentrations; however, no toxic effects were observed in the treated plants.

In agreement with the previous results, the 2D-PAGE protein gel blot analysis showed that 20 µM of CDPs treatment induced changes in *Zm*S6K electrophoretic mobility (Fig. 6). The changes may have resulted from post-translational modifications to which S6K is susceptible, mainly the addition of phosphate groups at multiple sites for complete S6K activation (*Fenton & Gout, 2011*). These results correlate with the increase in *Zm*S6K Thr-389 phosphorylation (Fig. 5), as well as the growth-promoting effect observed in maize plants treated with 20 µM CDPs for 8 and 30 days, suggesting an association between TOR kinase-mediated *Zm*S6K activation and the maize growth-promoting effects of the bacterial CDPs.

## CONCLUSIONS

The role of the TOR/S6K signaling pathway as the central regulator of cell growth and development in eukaryotes has been established since the past decade. Conservation of the TORC1 complex and up and downstream components have been revealed in plants, and currently there is a growing interest in the plant scientific community to clarify the steps of the signaling pathway involved in plant adaption processes regulated by it, to control plant growth and development. On the other hand, biotechnological strategies for

the development of products that increase the production of crops have been implemented, leading to the characterization of rhizospheric microorganisms, also as compounds produced by these microorganisms with plant growth promotion properties. *P. aeruginosa* is a ubiquitous bacterium that can be found colonizing plant roots causing pathogenesis; however, in our workgroup compounds able to improve the development of *A. thaliana* has been isolated from this bacterium, thus we now tested whether these compounds (CDPs) can have the same plant growth-promoting effect on maize, a cereal with economic interest worldwide. Our results showed that *P. aeruginosa* CDPs promote maize plant growth and development by modulating the root system architecture. The CDPs-induced promotion of growth and development in maize plants was correlated with increased levels of *Zm*S6K phosphorylation, suggesting the involvement of the TOR-S6K signaling pathway in the plant promoting effects of CDPs. This is the first study to show that plant–microorganism interaction phenomena mediated by organic molecules such as CDPs produced by members of the genus *Pseudomonas* involves the TOR/S6K signaling pathway as molecular signaling mechanism of plant growth promotion.

As previously discussed, it was established that the bacterial CDPs activate the auxin pathway in *A. thaliana*, however, the effects of CDPs on growth and development in this plant are different than those induced by auxinic compounds, opening the possibility that the activation of TOR signaling pathway in plants is also induced through alternative mechanisms. In our lab we are working on clarifying the CDPs action mechanism in plants, trying to find the molecular targets involved. As occurs in *Arabidopsis*, molecular docking studies could reveal some of the TOR pathway component involved in the CDPs interaction. In addition, with transcriptional studies we could establish whether the CDPs modify the expression of some plant TOR signaling pathway actors.

### Funding

This work was supported by the Coordinación de la Investigación Científica-Universidad Michoacana de San Nicolás de Hidalgo (CIC-UMSNH) program 2016-17, and by the Consejo Nacional de Ciencia y Tecnología (CONACyT)-México, grants number 222405 and number 256119. Iván Corona-Sánchez and Omar González-López were the recipients of CONACyT fellowships, numbers 584655 and 259020, respectively. The funders had no role in study design, data collection and analysis, decision to publish, or preparation of the manuscript.

### Grant Disclosures

The following grant information was disclosed by the authors:
Coordinación de la Investigación Científica-Universidad Michoacana de San Nicolás de Hidalgo (CIC-UMSNH) program 2016-17.
Consejo Nacional de Ciencia y Tecnología (CONACyT)-México: 222405 and 256119.
CONACyT fellowships: 584655 and 259020.

## Competing Interests

Jesus Campos-Garcia is an Academic Editor for *PeerJ*.

## Author Contributions

- Iván Corona-Sánchez performed the experiments, analyzed the data, prepared figures and/or tables, authored or reviewed drafts of the paper, approved the final draft.
- Cesar Arturo Peña-Uribe performed the experiments, analyzed the data, prepared figures and/or tables, authored or reviewed drafts of the paper, approved the final draft.
- Omar González-López performed the experiments, analyzed the data, prepared figures and/or tables, authored or reviewed drafts of the paper, approved the final draft.
- Javier Villegas conceived and designed the experiments, analyzed the data, contributed reagents/materials/analysis tools, authored or reviewed drafts of the paper, approved the final draft.
- Jesus Campos-Garcia conceived and designed the experiments, analyzed the data, contributed reagents/materials/analysis tools, authored or reviewed drafts of the paper, approved the final draft.
- Homero Reyes de la Cruz conceived and designed the experiments, analyzed the data, contributed reagents/materials/analysis tools, prepared figures and/or tables, approved the final draft.

## Data Availability

Raw figure data are available in the Supplemental Files.

## Supplemental Information

Supplemental information for this article can be found online at http://dx.doi.org/10.7717/peerj.7494#supplemental-information.

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
