# Peer review of "Cyclodipeptides from Pseudomonas aeruginosa modulate the maize (Zea mays L.) root system and promote S6 ribosomal protein kinase activation"

_PeerJ, doi:10.7717/peerj.7494_

## Round 0.1 · original submission · Major Revisions

Two reviewers raise substantial comments, that need to be addressed adequately.

The first one involves the curiosity that only one of the tested concentrations of CDPs has an effect, whereas one normally would expect a dose response curve with a local optimum, with adjacent concentrations also inducing a similar albeit less strong effect.
The second comment relates to the over interpretation of the data. To my mind also the major weakness of the manuscript. Although the results show a correlation between CDP treatment, growth and S6K activity, this does not prove S6K mediates downstream effects of CDPs.
Interaction with perturbations (genetic and/or chemical) between treatment and putative regulatory pathway would be required to substantiate this claim.

Besides these major concerns, a number of more minor issues considering the introduction and materials and methods also need to be addressed. I hope you will be able to address these issues in a next version of this interesting work.

Best Regards, Gerrit

Reviewer 1 ·

Basic reporting

The english language is generally good and the figures are well presented.

1. Regarding the introduction, the authors mainly cite a review (Dobrenel et al 2016) for TOR pathway in plants. More recent reviews or original papers should be cited to reflect the most recent advances in the field (for example, the sentence line 90-93 does not reflect the recent literature), particularly in relation to root development.

2. Line 72, Deprost 2005 does not report TOR gene KO (but raptor KO) so it is not relevant for this sentence.

3. Regarding the structure of the manuscript, the Results section is extremely short in comparison to other parts.

4. MAJOR CONCERN: More data should be included as supplementary informations. The authors write in the legend of figures 5 and 6 that “experiment was repeated twice with similar results”. As no statistical data is provided for these Western Blots, I recommend providing the second blots in supp data. Also, regarding Fig 5, a convincing rational for the selection of the cropped band with Phosphorylated S6K should be given as many bands of similar MW are observed on the full blot (Fig 5 raw in supp files). Idem for unphosphorylated SmS6K even if less bands are observed. These data and informations will help reviewers and readers analyzing the data.

Experimental design

This manuscript reports original primary research in the scope of PeerJ. Not ethical problems detected.

5. Methods are well described, but the sentence line 121-123 is not clear. Did they use the same amount of DMSO for each condition or not?

Validity of the findings

6. MAJOR CONCERN: In Figure 2, the authors show the effect of 5 doses of CDPs on maize seedlings. That is nice to show such different doses, but surprisingly, no dose dependent effects were observed and CDPs affected root growth only at the concentration of 20 µM. This absence of a dose-dependence suggests that the effect observed might be an artefact. Testing other doses could help resolve this issue.

7. MAJOR CONCERN: As written in the discussion (lines 283-286), this study reports a correlation between CDP effect of root growth and ZmS6K modification. The association between S6K modification and growth promotion is only suggested by this correlation. Therefore, the title “by a mechanism involving S6 ribosomal protein kinase activation” is not supported by the data and must be changed. An alternative to modification of the title would be to provide experimental evidences supporting the claim of the title.

8. Results line 207-208: “All the parameters showed increasing trends in CDP-treated plant (Fig 4)”. This in not correct for the parameter in Fig 4D that showed a slightly decreasing trend.

Additional comments

In sections 1 to 3, my most important comments are indicated by “MAJOR CONCERN”. They must be addressed prior to publication as they relate to the possibility to fully appreciate the data (4), the robustness of the findings (6) and the Title’s statement (7)

Reviewer 2 ·

Basic reporting

No comment

Experimental design

Regarding the cyclodipeptides (CDPs) used in this study, authors refer to Ortiz-Castro et al. (2011), who isolated and identified three CDPs ( cyclo(L-Pro-L-Tyr), cyclo(L-Pro-L-Val), and cyclo(L-Pro-L-Phe). First, autjors should show include mass spectra of purified CDPs fractions and also indicate if they used a particular CDPs of a mixture of the three isolated CDPs.

Authors should also clarify the selection of used concentrations should be given, e.g.,12 uM of auxin.

Validity of the findings

In this study, authors showed that cyclodipeptides (CDPs) promote the growth of maize plant by modulating the root system architecture. The CDPs treatment elicited a dose-dependent increase in ZmS6K phosphorylation, suggesting the involvement of the TOR-S6K signaling pathway in the growth promoting effects of CDPs.
Although the study indicates a correlation between CDPs treatment and S6K phosphorylation, more analyses are needed to validate the involvement of TOR-S6K signaling in root growth promotion. In this regards, authors should test if the CDPs treatment increases TOR kinase and the ZmS6K activation resulted in S6 ribosomal phosphorylation

The activity of plant S6Ks has been shown to increase in response to auxin, also CDPs require a canonical auxin signaling pathway for activity (Ortiz-Castro et al. 2011). Thus CDPs induces S6K activity may be auxin dependent, please discuss it.

Annotated reviews are not available for download in order to protect the identity of reviewers who chose to remain anonymous.

---

## Round 0.2 · Minor Revisions

It appears that many strides were taken to get to the newest revision of the manuscript. There was some concern about the validity of one of the assays due to omission of important control data. I will place this at a ‘minor revision’ level until the item in question can be resolved or rebutted to an acceptable degree as I regard ths to be a valid reservation. I was surprised at the rather short conclusion statement; I would think that the authors would have more to say. Perhaps in the interim the authors can reflect on other statements that may be made to sway readers in accepting the claimed role, or at least atone for what may be needed to further confirm their findings. I can see the association, but it would be beneficial for additional information to be added regarding the signaling pathway so that a clear network of the pathway as a plan for future study may be presented to pinpoint a target role. Thank you for your progress, and we look forward to the enhanced revision.

Reviewer 1 ·

Basic reporting

4B. In response to my MAJOR CONCERN 4, the authors have provided as supplementary information (Fig5wb2raw.png) the second blot western blot for p-ZmS6K. However, they did not provide the corresponding Western blot for unphosphorylated S6K that was requested during the first reviewing. It is therefore not possible to claim that “The experiment was repeated twice with similar results” (legend Figure 5). Indeed, Fig5wb2raw.png suggests different results than Fig 5, particularly regarding line 2 and 7. This might be due to differences of amount of protein loaded or S6K amount, hence the importance of the unphosphorylated S6K control.

If the authors have difficulties with the anti-human-phospho-p70S6K-Thr389 antibody, they might have better results with a plant specific antibody like ab207399.


5B. In response to my concern 5, the authors should simply write in the manuscript the amount of DMSO used.

Experimental design

no comments

Validity of the findings

no comments

Additional comments

The authors have properly responded to my comments exept for comments 4 and 5.

Reviewer 2 ·

Basic reporting

no comment

Experimental design

no comment

Validity of the findings

no comment

Additional comments

The authors made amendments that considerably improved the manuscript, and I consider that it is OK for publication.

---

## Round 0.3 · accepted · Accept

The authors have responded to comments made by reviewers and have made statements resolving their observed findings. It was shown that microbial derived CPDs had an influence on the growth patterns of maize roots that were measured in a dose-response, and that phosphorylation of maize S6K was related to the dose-response pattern suggesting that the TOR/S6K signaling pathway may be key in this interaction. The information presented here may be of value for establishing other roles for which microorganisms influence plant growth. The manuscript should be ready for publication and is considered accepted. Thank you for your contribution.